# Pectoralis minor length index at 1 month postoperative can predict homolateral neuropathic pain 4 months after mastectomy with lymph node resection

Asall Kim[1], Chunghwi Yi[2], Myungki Ji[1], Ui-Jae Hwang[3], Jae-Young Lim[4], Yujin Myung[5], Eun Joo Choi[6], Hee-Chul Shin[7], Jaewon Beom[4]*

1 Department of Rehabilitation Medicine, Seoul National University Bundang Hospital, Seongnam, Gyeonggi-do, Republic of Korea, 2 Department of Physical Therapy, College of Software and Digital Healthcare Convergence, Yonsei University, Wonju, Gangwon-do, Republic of Korea, 3 Department of Rehabilitation Sciences, The Hong Kong Polytechnic University, Hung Hom, Hong Kong (SAR), China, 4 Department of Rehabilitation Medicine, Seoul National University College of Medicine, Seoul National University Bundang Hospital, Seongnam, Gyeonggi-do, Republic of Korea, 5 Department of Plastic and Reconstructive Surgery, Seoul National University College of Medicine, Seoul National University Bundang Hospital, Seongnam, Gyeonggi-do, Republic of Korea, 6 Department of Anesthesiology and Pain Medicine, Seoul National University College of Medicine, Seoul National University Bundang Hospital, Seongnam, Gyeonggi-do, Republic of Korea, 7 Department of Surgery, Seoul National University College of Medicine, Seoul National University Bundang Hospital, Seongnam, Gyeonggi-do, Republic of Korea

* powe5@snubh.org

## Abstract

The relationship between postoperative physical changes and the development of homolateral neuropathic pain (HLNP) following mastectomy and lymph node resection remains poorly understood. In this study, we aimed to investigate whether early postoperative physical and symptom-based assessments could predict HLNP occurrence at 4 months post-surgery. Fifty-seven breast cancer survivors were included, with HLNP defined as a painDETECT Questionnaire score ≥ 13 at 4 months. Independent variables included patient demographics, physical function metrics including pectoralis minor length index (PMI), and questionnaire-based evaluations at 1 month postoperatively. Multivariate logistic regression identified systemic therapy side effects (ST) (odds ratio [OR]: 1.056; 95% confidence interval [CI]: 1.015–1.098) and PMI (OR: 0.204; 95% CI: 0.043–0.977) as significant predictors of HLNP. Receiver operating characteristic curve analysis identified cutoff values of 23.81 for ST and 9.82 for PMI. Reconstruction type and adjuvant therapy influenced the correlation between PMI and the number of resected lymph nodes, unlike external rotation metrics. Early assessment of ST and PMI facilitates HLNP risk prediction following breast cancer surgery. Multimodal interventions, including targeted physical therapy, may mitigate HLNP risk, highlighting the importance of early postoperative care.

**Data availability statement:** All relevant data are within the manuscript and its Supporting Information files. (Dataset are also available from the zenodo site (DOI: 10.5281/zenodo.14266619)).

**Funding:** The author(s) received no specific funding for this work.

**Competing interests:** The authors have declared that no competing interests exist.

## Introduction

With 5-year survival rates for breast cancer now exceeding 90% [1,2], restoring upper limb function is crucial for breast cancer survivors (BCS). Recent studies emphasize the significant role of homolateral neuropathic pain (HLNP), localized near the breast surgery site, in upper limb dysfunction [3]. The International Association for the Study of Pain defines neuropathic pain as "pain arising from a direct consequence of a lesion or disease of the somatosensory system" [4]. Despite the importance of early detection [5], HLNP management remains challenging due to complex underlying mechanisms and the lack of standardized diagnostic criteria [6].

Several screening tools have been designed to identify neuropathic pain [7]. Studies utilizing validated tools report various predictors of postsurgical pain in BCS within the first postoperative year, including demographic factors (e.g., body mass index, age, diabetes), psychological factors (e.g., anxiety), type of surgical procedure (e.g., axillary clearance, volume of resected breast tissue), cancer treatment type, and postoperative symptoms (e.g., pain and arm symptoms) [8–15]. These factors may directly or indirectly affect primary afferent fibers, triggering peripheral and central sensitization and ultimately altering the pain modulation system [5,16].

However, the role of pectoral muscle dysfunction in the development of neuropathic pain among BCS remains underexplored. Specifically, pectoralis minor muscle tightness may contribute to HLNP development through its impact on surrounding tissues. Pectoralis minor muscle tightness alters scapular biomechanics [17,18] and reduces upper limb range of motion (ROM) [19,20]. Consequently, repetitive movements with abnormal kinetic patterns can result in musculoskeletal conditions such as rotator cuff tears [21]. Lee et al. [20] identified chest tightness, assessed via pectoralis minor muscle length, as a contributing factor to upper limb pain after breast cancer surgery. From a pain perspective, previous studies have reported that postoperative pain may induce muscle hypertonicity [22]. Altered muscle activation patterns caused by anterior chest wall tightness may reflect protective responses to pain arising from irritation of injured tissues. Furthermore, scapular depression has been linked to increased neural sensitivity [23] and possible compression or entrapment of nerves and blood vessels in the axillary region [24]. Taken together, these findings suggest that pectoralis minor tightness may be a consequence of both tissue damage and a contributing factor to neural irritation. Thus, we hypothesize that pectoralis minor tightness could serve as an indicator of HLNP, as the tightness may either cause postoperative pain or result from it.

Lymph node resection, a common procedure in breast cancer surgery, may influence postoperative musculoskeletal changes. Yang et al. [21] suggested that postoperative pain or radiation therapy could lead to pectoral muscle hypertonicity or fibrosis, no study has directly examined whether lymph node resection contributes to pectoralis minor muscle shortening.

Previous research has reported a marginally statistically significant negative correlation between shoulder external rotation range and the number of lymph nodes resected [25], suggesting that lymph node removal may impact musculoskeletal function. Given the anatomical proximity of lymphatic tissue to the pectoralis minor muscle [26], extensive lymph node resection may influence muscle length and consequently reduce upper limb ROM.

Therefore, in this study, we aimed to investigate whether postoperative pectoralis minor muscle changes significantly predict neuropathic pain development following breast cancer surgery Furthermore, we hypothesized that the number of resected lymph nodes would correlate with postoperative musculoskeletal alterations. These findings may improve clinical management strategies for postoperative pain and help mitigate long-term upper limb dysfunction in BCS.

## Materials and methods

### Participants

This study analyzed data from a prospective cohort study [27] conducted in accordance with the Strengthening the Reporting of Observational Studies in Epidemiology guidelines and the Declaration of Helsinki. Ethical approval was obtained from the Institutional Review Board of Seoul National University Bundang Hospital (IRB number: B-2108-702-309), and all participants provided written informed consent. The study was registered with the Clinical Research Information Service prior to participant enrollment (Trial registration number: KCT 0006501).

The inclusion criteria were as follows: (1) BCS aged ≥18 years and (2) patients who underwent mastectomy with immediate breast reconstruction (IBR) [27]. Given that IBR is typically recommended for younger patients, no upper age limit was established. These patients were selected because their standard postoperative care included a follow-up visit at the rehabilitation medicine outpatient clinic approximately 1 month after surgery. Eligible patients were informed about the study during clinic visits and provided sufficient information to decide on participation. Exclusion criteria encompassed: (1) bilateral mastectomy, (2) history of previous breast cancer surgery, (3) stage IV cancer, (4) declining to enroll in the study, (5) requiring acute medical care due to complications such as surgical site inflammation, severe lymphedema, or upper limb condition (e.g., rotator cuff disease or axillary web syndrome) [27], (6) diabetes mellitus (to exclude confounding by diabetic neuropathy), and (7) no lymph node resection (sentinel lymph node biopsy or axillary lymph node dissection), as these patients were excluded from correlation analysis to avoid outliers.

### Procedure

Participants were recruited for this prospective study between August 24, 2021 and February 27, 2022. Following the screening examination, the first evaluation was conducted 1 month postoperatively (within 1 week of screening), and the second evaluation occurred 4 months postoperatively. During the screening examination, demographic data, surgical details, treatment plans, and chemotherapy regimens were extracted, and the history of adjuvant therapy was confirmed from electronic medical records. During both evaluations, patients underwent physical examinations and completed questionnaire-based assessments. The number of resected lymph nodes was obtained from the electronic medical records. Participants were scheduled for physical therapy sessions. A.K. provided education and printed materials on pectoralis minor muscle stretching and scapular retraction exercises to all participants during the first educational session (S1 Appendix). Among the participants, 36 patients attended physical therapy for a median of three extra sessions (interquartile range [IQR]: 2–4 sessions), per the prescribed treatment plan. During these additional sessions, A.K. guided patients through shoulder elevation ROM exercises and administered deep frictional massage around the pectoralis minor muscles. Furthermore, 13 participants were prescribed self-care education sessions focused on edematous management, with a median of three sessions (IQR: 1–4 sessions), all conducted by a specialized physical therapist. Patients did not receive physical therapy such as thermoelectrical modality, to reduce pain. The overall study flow is presented in Fig 1.

### Outcome measures

This study employed both physical examinations and questionnaire-based measurements. The painDETECT questionnaire (PDQ) was conducted to evaluate HLNP, which was a dependent variable of the study, while independent variables included pectoral muscle length index (PMI), ROM, quality of life, functional symptoms, and upper limb disability.

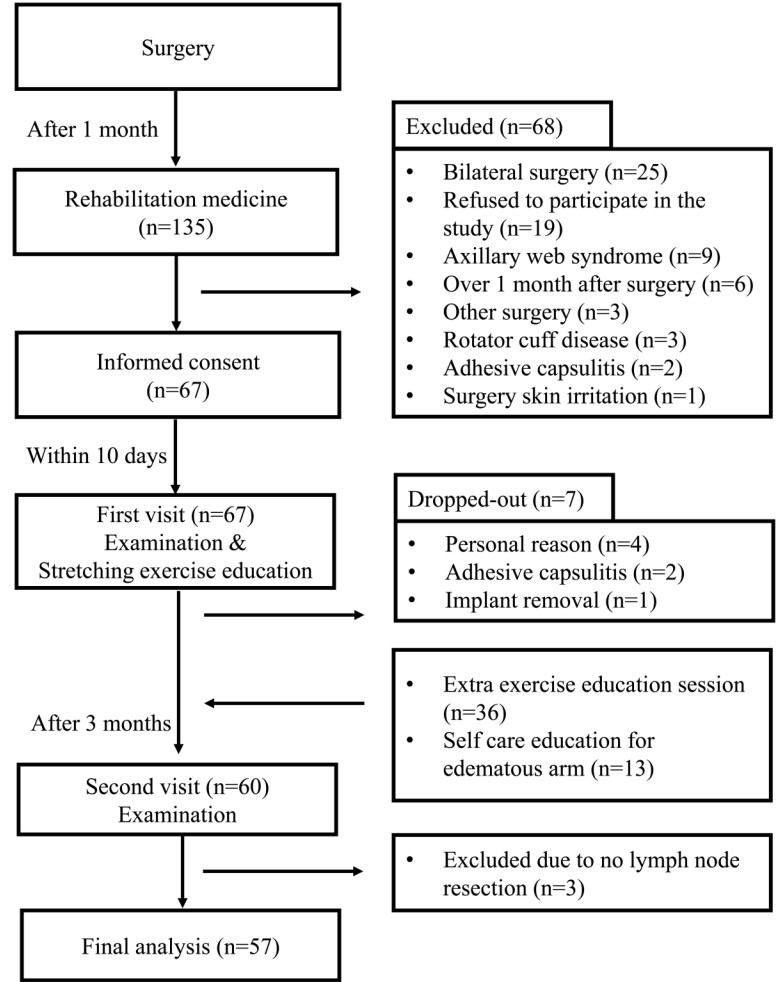

**Fig 1. Study flow diagram.**

**Dependent variable.** The presence and intensity of HLNP in the axillary and breast surgical regions were assessed using a validated painDETECT questionnaire (PDQ) developed based on neuropathic pain characteristics [28]. The PDQ is a validated tool for detecting neuropathic pain [7] and is widely used to identify neuropathic pain predictors in BCS [6,11]. It consists of four parts: the first assesses pain intensity, while the remaining three parts evaluate pain location and characteristics. The total score is derived from the sum of the last three parts [28]. The PDQ demonstrates excellent discriminative validity, with established cutoff values: a score of 0–12 points indicates neuropathic pain is unlikely (< 15%), a score of 13–18 suggests neuropathic pain cannot be ruled out, and a score ≥19 indicates a high probability of neuropathic pain (> 90%) [28].

The original English version of the painDETECT questionnaire demonstrates a sensitivity and specificity of 85% and 80%, respectively, when differentiating neuropathic from nociceptive pain in patients with chronic low back pain. The Korean PDQ version used in our study has a sensitivity of 95.4% and a specificity of 73.8%, 13 points or lower to indicate nociceptive pain, with ≥19 points having a sensitivity and specificity of 95.4% and 73.8%, respectively, for neuropathic pain [29].

Considering that chronic pain is defined as pain lasting ≥ 3 months [30], HLNP was identified 4 months postoperatively. For primary analyses, HLNP was defined as a PDQ score ≥13, effectively distinguishing neuropathic pain from nociceptive

pain [28]. Furthermore, HLNP was defined as a PDQ score ≥ 19 employed to differentiate neuropathic pain from both nociceptive and ambiguous pain.

**Independent variables.** The pectoralis minor length index (PMI) is a key independent variable in this study, quantifying pectoralis minor muscle tightness through standardized length measurements. While various methods and positions have been used to examine pectoralis minor muscle length in BCS [20,31], this study measured the distance between the fourth intercostal space near the sternum and the coracoid process was measured in the supine position [20,32,33]. This position was selected to minimize the effect of gravity on posture [33] and was measured using a Palpation Meter (Performance Attainment Associates, St. Paul, MN, USA; Fig 2). Intra-rater reliability of the measurement in supine position was between 0.87 and 0.93 in patients with and without shoulder impingement symptoms [32]. The intra-tester reliability of the Palpation Meter was reported as ICC = 0.971, and it demonstrated good validity when compared with motion capture systems (r = 0.87) [31]. The measured length was standardized by participant height to calculate PMI using the formula: PMI = [(pectoralis minor muscle length/height) × 100] [34,35]. Based on established criteria, a PMI value of 10 was considered indicative of normal resting muscle length in young, asymptomatic individuals [33]. However, to date, there is no published literature reporting a normative PMI value in BCS. To ensure measurement precision, three repeated assessments were conducted for each participant, with the average value used for subsequent analyses.

Active ROM was evaluated for shoulder flexion, abduction, and internal and external rotation using a validated mobile application [36]. A mobile phone was attached to the operated arm using an armband. Participants were asked to raise the arm in sagittal and frontal plane in a sitting position to examine active ROM for flexion and abduction, respectively. Subsequently, participants were asked to lie on the examination bed, with their arms abducted to 45˚ and the elbow flexed to 90°, while the mobile phone remained attached to lower arm during external and internal arm rotation. The reliability of ROM assessments was > 0.9 [27]. Three repetitions were conducted, and average data was used in the study. Pectoral tightness was defined as an average forward flexion and abduction < 170°, combined with external rotation > 80° [21], and tightness of the pectoralis major muscle.

Pain intensity was measured using the 11-point numeric rating scale (0–10) embedded within the PDQ. Participants reported their current pain intensity, strongest pain experienced in the last 4 weeks, and the average pain during the same period. Quality of life, functions, and symptoms were assessed using the Korean versions of the European Organization for Research and

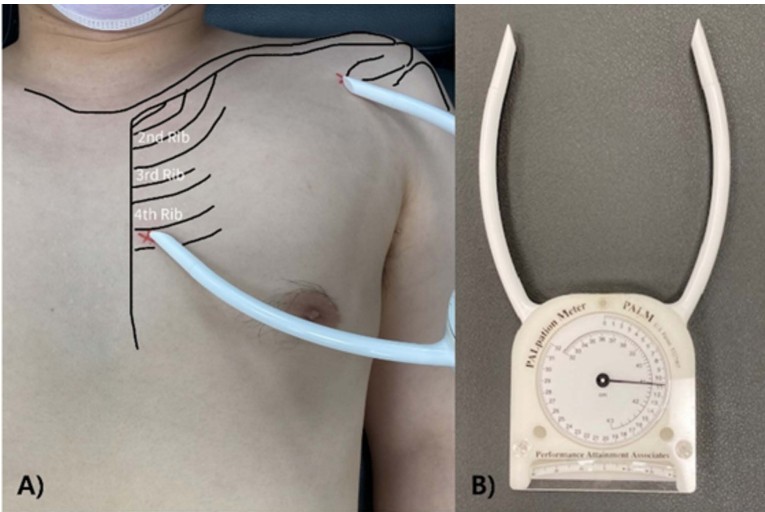

**Fig 2. Measurement of pectoralis minor muscle length.** (A) Pectoralis minor muscle length was measured in a supine lying position. (B) Palpation Meter.

Treatment of Cancer Quality of Life Questionnaire Core 30 and the breast cancer-specific questionnaires [37,38], following their respective manuals [39,40]. Symptom scales included fatigue, insomnia, pain, arm and breast symptoms, body image, and systemic therapy side effects (ST). Each symptom was evaluated through multiple items [39,40]. For instance, the ST scale includes seven items assessing symptoms such as dry mouth, altered taste, eye pain, hair loss, general illness, hot flashes, and headaches. Both questionnaires demonstrated good validity and reliability in BCS [41]. Upper limb disability was measured using the Quick Disabilities of the Arm, Shoulder, and Hand (Quick-DASH) questionnaire [42], which has demonstrated acceptable validity and high level of reliability in BCS [43], and responsiveness in upper limb patients [44]. Studies reported a Cronbach's alpha of 0.93 and test-retest reliability of 0.78, and this tool was validated to use in BCS. The Korean version of the questionnaire was used [45], and permission to use all questionnaires was obtained from their respective official sources.

## Statistical analysis

All statistical analyses were performed using SPSS version 26 (IBM Corp., Armonk, NY, USA), with statistical significance set at $p < 0.05$. The assumption of normal distribution was assessed using the Shapiro–Wilk test. Categorical data are presented as frequencies (percentages), and continuous variables as mean (standard deviation). Group comparisons between BCS with PDQ scores ≥13 and < 13 were conducted based on data distribution normality.

Given the relatively small sample size, a two-step logistic regression analysis was conducted to identify predictors. First, univariate logistic regression analyzed independent variables including age, body mass index, postmenopausal status, type of lymph node surgery, type of reconstruction, type of adjuvant therapy, pain medication during follow-up, and results from physical and questionnaire-based examinations. Variables with a p-value < 0.15 in the univariate analysis were included in forward multivariate logistic regression. Odds ratios (OR) and 95% confidence intervals (CI) were quantified for each independent variable. The model fit was assessed using a chi-square test and its classification accuracy. The Hosmer–Lemeshow test evaluated model fit, with $p > 0.05$ indicating a better fit.

Sensitivity analysis was performed under two conditions: (1) incorporating time since surgery to enrollment as a covariate for multivariate analysis, accounting for its impact on physical recovery (ranging from 26 to 57 days due to the study's clinic schedule), and (2) applying an alternative HLNP definition (PDQ score ≥19 [11]). The PDQ ≥ 19 threshold reliably distinguishes definitive neuropathic pain from nociceptive/ambiguous pain and has been validated in previous BCS pain studies [6,11]. Both the two-step logistic regression and covariate-controlled multivariate regression were repeated for this alternative definition.

Receiving operating characteristic (ROC) analysis was performed using MedCalc® Statistical Software version 23.0.2 (MedCalc Software Ltd, Ostend, Belgium; https://www.medcalc.org; 2024) to establish cut-off values for identified predictors. An area under the curve (AUC) > 0.7 was considered acceptable [46]. A one-tailed Spearman's rank test assessed correlations between the amount of resected lymph nodes and physical examination results, with coefficients < 0.4 considered weak and > 0.7 considered strong [47].

Sample size calculations and post-hoc power analysis were conducted using G*power software (latest version 3.1.9.7; Heinrich-Heine-Universität Düsseldorf, Düsseldorf, Germany). The power analysis determined that at least 42 participants would be required (OR: 3 [11]; α = 0.05; power = 0.8; and probability of Y: 0.5 [6]). Therefore, the final sample size was sufficient to run this analysis.

## Results

Among 135 screened patients, 60 completed follow-up evaluations [27], with 57 ultimately included in the final analysis after excluding three individuals who did not undergo lymph node resection (Table 1). Of the 75 excluded patients, 19 declined participation while the remaining 56 were excluded for medical reasons (Fig 1).

Surgical records confirmed no muscle or nerve injuries. Among the 60 evaluated patients, 18 received pain medication during follow-up. Seventeen patients were prescribed nonsteroidal anti-inflammatory drugs, including 15 who received aceclofenac as part of their chemotherapy regimen and two who were prescribed celecoxib at the Department

**Table 1. Baseline characteristics of patients.**

| | N = 57[a] |
|---|---|
| Age (years) | 45.7 ± 6.9 |
| Body mass index (kg/m$^2$) | 22.3 ± 2.6 |
| Post-menopause at enrollment | 7 (12) |
| Tumor stage | |
| In situ/ 1/ 2/ 3 | 7 (12)/ 28 (49)/ 19 (33)/ 3 (5) |
| Node stage | |
| 0/ 1/ 2/ 3 | 40 (70)/ 12 (21)/ 3 (5)/ 2 (4) |
| Type of mastectomy | |
| Nipple-sparing/ Skin-sparing/ Total mastectomy | 44 (77)/ 9 (16)/ 4 (7) |
| Type of immediate breast reconstruction | |
| Abdominally-/Implant-based breast reconstruction | 25 (44)/ 32 (56) |
| Type of lymph surgery | |
| SLNB/ALND/Both | 44 (77)/ 5 (9)/ 8 (14) |
| Number of lymph nodes resected | |
| All (n = 57) | 7.5 ± 7.8 |
| SLNB (n = 44) | 3.9 ± 2.1 |
| ALND (n = 5) | 18.8 ± 11.1 |
| Both (n = 8) | 20.0 ± 6.4 |
| Surgery on the dominant hand side | 30 (53) |
| Adjuvant therapy | |
| Chemotherapy | 29 (51) |
| Regimen of chemotherapy (TC/AC/ACT) | 17/ 4/ 8 |
| Radiation therapy | 16 (28) |
| Hormone therapy | 37 (65) |
| Tamoxifen only/ Tamoxifen + goserelin | 32 (56)/ 5 (9) |

[a]Characteristics were reported as mean ± SD for continuous variables and as frequencies (%) for categorical variables. SLNB, sentinel lymph node biopsy; ALND, axillary lymph node dissection; TC, docetaxel + cyclophosphamide; AC, doxorubicin + cyclophosphamide; ACT, doxorubicin + cyclophosphamide + docetaxel.

of Hematology and Oncology Clinics. Of the aceclofenac recipients, two patients were also prescribed gabapentin and tramadol, respectively at the Department of Plastic Surgery Clinic 1 month postoperatively. Furthermore, one patient received only gabapentin at the Department of Plastic Surgery Clinic 1 month postoperatively.

At 1 month postoperatively, 27 patients (47%) were categorized as unlikely to have NP, 24 (42%) as having ambiguous NP, and 6 (11%) as highly likely to have NP. At 4 months, these proportions shifted to 61%, 25%, and 14%, respectively. By the 4-month mark, 22 patients (39%) met the criteria for HLNP. Within-group differences over time were not statistically significant (p = 0.062); however, comparisons between the HLNP and non-HLNP groups revealed significant disparities, particularly in PMI, which was higher in the non-HLNP group (p = 0.012) (Table 2). Questionnaire-based assessments indicated that patients with HLNP reported more severe Quick-DASH, ST, and arm and breast symptoms. Among these, ST exhibited the only statistically significant difference between groups (p = 0.005).

## Factors associated with HLNP

Univariate logistic regression analysis identified 15 significant predictors of HLNP (PDQ score ≥ 13) (Table 3). The most significant predictor was ST, with an OR of 1.058 (95% CI: 1.017–1.100, *p* = 0.005). We consider the role of this predictor

**Table 2. Physical and questionnaire-based evaluations at 1 month postoperative in patients with and without homolateral neuropathic pain at 4 months postoperative.**

| Variables | All (n = 57) | Non-HLNP (n = 35) | HLNP (n = 22) | p-value |
|---|---|---|---|---|
| Pectoral tightness (yes)[a] | 36 (63) | 24 (69) | 16 (73) | 0.738 |
| PMI[b] | 9.93 (0.44) | 10.05 (0.43) | 9.75 (0.41) | 0.012* |
| Shoulder range of motion (degree) | | | | |
| Flexion | 142.20 (27.99) | 142.2 (22.07) | 143.59 (33.70) | 0.544 |
| Abduction | 144.40 (41.12) | 148.10 (38.00) | 132.19 (65.32) | 0.309 |
| External rotation | 84.87 (15.91) | 85.93 (12.44) | 81.95 (26.74) | 0.265 |
| Internal rotation | 83.10 (11.25) | 82.73 (11.27) | 84.17 (9.72) | 0.350 |
| Quick–DASH score | 25.00 (21.59) | 25.00 (20.45) | 32.96 (21.02) | 0.094 |
| painDETECT questionnaire | | | | |
| Present pain intensity | 2 (3) | 2 (3) | 2.5 (2) | 0.109 |
| Strongest pain intensity last 1 month | 4 (4) | 3 (4) | 5.5 (4) | 0.038* |
| Average pain intensity last 1 month | 3 (2) | 3 (2) | 3 (3) | 0.098 |
| EORTC QLQ C-30/BR-23 | | | | |
| Quality of life | 59.65 (20.16) | 63.09 (15.30) | 54.17 (25.56) | 0.149 |
| Physical function | 72.86 (14.79) | 75.24 (12.99) | 69.09 (16.91) | 0.128 |
| Role function | 66.67 (33.33) | 66.67 (33.33) | 66.67 (33.33) | 0.542 |
| Emotional function | 75.00 (25.00) | 83.33 (25.00) | 75.00 (27.08) | 0.176 |
| Social function | 66.67 (33.33) | 66.67 (33.33) | 66.67 (37.51) | 0.184 |
| Fatigue | 33.33 (27.79) | 33.33 (22.23) | 33.33 (33.34) | 0.846 |
| Pain | 33.33 (16.66) | 33.33 (16.66) | 33.33 (33.33) | 0.484 |
| Insomnia | 33.33 (50.01) | 33.33 (66.67) | 33.33 (33.34) | 0.386 |
| Systemic therapy side effect | 19.05 (23.81) | 14.29 (14.29) | 33.33 (29.76) | 0.005** |
| Arm symptoms | 33.33 (22.22) | 22.22 (11.11) | 33.33 (22.22) | 0.054 |
| Breast symptoms | 25.00 (25.00) | 16.67 (25.00) | 25.00 (18.74) | 0.070 |
| Body image | 66.67 (45.83) | 66.67 (33.33) | 54.17 (56.25) | 0.259 |

[a]Data expressed as frequencies (%) and chi-square test was used;

[b]data described as mean (SD) and the independent *t*-test was conducted; Otherwise, data described as median (IQR) and the Mann-Whitney U test was performed. HLNP, homolateral neuropathic pain; PMI, pectoralis minor length index; Quick–DASH, Quick Disability of the arm, shoulder, and hands; PDQ, painDETECT questionnaire.

*and

**indicates p-value < 0.05 and < 0.01, respectively.

or preceding factor for HLNP prevalence as clinically significant; however, further interpretation requires caution on the modest OR effect size. Other predictors included the PMI (OR: 0.185; 95% CI: 0.046–0.745, p = 0.018), the strongest pain intensity during the past 4 weeks (OR: 1.311; 95% CI: 1.023–1.681, p = 0.032), the average pain intensity over the past 4 weeks (OR: 1.390; 95% CI: 1.010–1.915, p = 0.044), current pain intensity (OR: 1.369; 95% CI: 0.982–1.908, p = 0.064), quality of life (OR: 0.977; 95% CI: 0.950–1.005, p = 0.110), external rotation (OR: 0.968; 95% CI: 0.931–1.007, p = 0.111), arm symptoms (OR: 1.028; 95% CI: 0.993–1.064, p = 0.115), breast symptoms (OR: 1.025; 95% CI: 0.993–1.057, p = 0.129), pain medication during follow-up (OR: 1.975; 95% CI: 0.818–4.767, p = 0.130), physical function (OR: 0.971; 95% CI: 0.935–1.009, p = 0.131), Quick-DASH score (OR: 1.026; 95% CI: 0.991–1.061, p = 0.145), body mass index (OR: 0.844; 95% CI: 0.670–1.062, p = 0.147), abduction (OR: 0.987; 95% CI: 0.969–1.005, p = 0.149), and emotional function (OR: 0.976; 95% CI: 0.944–1.009, p = 0.149). When HLNP was redefined as a PDQ score ≥ 19, PMI approached statistical significance (OR: 0.177, 95% CI: 0.029–1.095, p = 0.062), followed by current pain intensity (OR: 1.408; 95% CI:

**Table 3. Factors associated with homolateral neuropathic pain according to different cutoff values (PDQ score ≥ 13 and ≥ 19).**

| Dependent variables | Independent variables | | | |
|---|---|---|---|---|
| | PDQ ≥ 13 | | PDQ ≥ 19 | |
| | *p*-value | EXP(b) (95% CI) | *p*-value | EXP(b) (95% CI) |
| Age | 0.655 | 0.982 (0.906–1.064) | 0.263 | 0.928 (0.814–1.058) |
| Body mass index | 0.147* | 0.844 (0.670–1.062) | 0.179 | 0.778 (0.539–1.122) |
| Post-menopause | 0.805 | 1.224 (0.247–6.074) | 0.984 | 1.024 (0.107–9.838) |
| T stage | 0.460 | 1.309 (0.640–2.677) | 0.218 | 1.895 (0.685–5.237) |
| N stage | 0.531 | 1.252 (0.620–2.527) | 0.414 | 1.434 (0.604–3.402) |
| Implant-based breast reconstruction | 0.367 | 1.653 (0.554–4.929) | 0.697 | 1.358 (0.292–6.322) |
| Axillary dissection | 0.991 | 0.993 (0.278–3.540) | 0.873 | 1.152 (0.203–6.530) |
| Surgery on the dominant side | 0.819 | 0.882 (0.303–2.571) | 0.362 | 2.045 (0.439–9.523) |
| History of chemotherapy | 0.327 | 1.715 (0.583–5.047) | 0.158 | 3.391 (0.622–18.486) |
| History of radiation therapy | 0.618 | 1.348 (0.417–4.363) | 0.150* | 3.083 (0.667–14.257) |
| History of hormone therapy | 0.682 | 1.266 (0.409–3.917) | 0.179 | 4.433 (0.505–38.932) |
| Pain medication during follow-up | 0.130* | 1.975 (0.818–4.767) | 0.574 | 1.380 (0.449–4.242) |
| Pectoral tightness (y/n)[a] | 0.739 | 1.222 (0.376–3.973) | 0.998 | $4.04 \times 10^8$ (not estimable) |
| Pectoralis minor length index[a] | 0.018* | 0.185 (0.046–0.745) | 0.062* | 0.177 (0.029–1.095) |
| Shoulder range of motion[a] | | | | |
| Flexion | 0.348 | 0.989 (0.965–1.013) | 0.602 | 0.992 (0.960–1.024) |
| Abduction | 0.149* | 0.987 (0.969–1.005) | 0.801 | 0.997 (0.972–1.022) |
| External rotation | 0.111* | 0.968 (0.931–1.007) | 0.202 | 1.052 (0.973–1.136) |
| Internal rotation | 0.668 | 1.010 (0.963–1.060) | 0.610 | 1.021 (0.942–1.108) |
| Quick-DASH score[a] | 0.145* | 1.026 (0.991–1.061) | 0.182 | 1.031 (0.986–1.079) |
| Pain intensity[a] | | | | |
| Present pain intensity | 0.064* | 1.369 (0.982–1.908) | 0.103* | 1.408 (0.933–2.125) |
| Strongest pain intensity last 1 month | 0.032* | 1.311 (1.023–1.681) | 0.470 | 1.126 (0.816–1.554) |
| Average pain intensity last 1 month | 0.044* | 1.390 (1.010–1.915) | 0.224 | 1.290 (0.855–1.947) |
| EORTC QLQ C-30/BR-23[a] | | | | |
| Quality of life | 0.110* | 0.977 (0.950–1.005) | 0.719 | 0.993 (0.957–1.030) |
| Physical function | 0.131* | 0.971 (0.935–1.009) | 0.803 | 0.994 (0.945–1.045) |
| Role function | 0.471 | 0.991 (0.968–1.015) | 0.602 | 0.991 (0.960–1.024) |
| Emotional function | 0.149* | 0.976 (0.944–1.009) | 0.984 | 1.000 (0.956–1.045) |
| Social function | 0.240 | 0.988 (0.969–1.008) | 0.546 | 0.992 (0.966–1.019) |
| Fatigue | 0.601 | 1.008 (0.979–1.036) | 0.934 | 1.002 (0.963–1.042) |
| Pain | 0.284 | 1.015 (0.988–1.042) | 0.106* | 1.029 (0.994–1.065) |
| Insomnia | 0.423 | 1.008 (0.989–1.027) | 0.907 | 0.998 (0.972–1.025) |
| Systemic therapy side effect | 0.005* | 1.058 (1.017–1.100) | 0.293 | 1.021 (0.982–1.060) |
| Arm symptoms | 0.115* | 1.028 (0.993–1.064) | 0.158 | 1.031 (0.988–1.075) |
| Breast symptoms | 0.129* | 1.025 (0.993–1.057) | 0.663 | 1.009 (0.969–1.050) |
| Body image | 0.228 | 0.989 (0.971–1.007) | 0.160 | 0.982 (0.959–1.007) |

HLNP, homolateral neuropathic pain near the surgery area at 4 months postoperatively; Quick–DASH, Quick Disability of the arm, shoulder, and hands; PDQ, painDETECT Questionnaire score.

[a]Indicates variables measured at 1 month postoperative,

*Indicates variables included in multivariate logistic regression (*p* < 0.15).

0.993–2.125, p = 0.103), pain subscale from the breast cancer-specific questionnaire (OR: 1.029; 95% CI: 0.994–1.065, p = 0.106), and radiation therapy history (OR: 3.083; 95% CI: 0.667–14.257, p = 0.064). PMI and current pain intensity exceeded the statistical threshold in both univariate regression analyses. While current pain intensity showed consistent estimates (OR: 1.4; CI: 0.9–2.0), PMI's wide confidence interval warrants careful interpretation.

A multivariate logistic regression model incorporated ST, and PMI achieved statistical significance (Cox & Snell $R^2$ = 0.238, $\chi^2$ = 15.466; p < 0.001) with a classification accuracy of 82.5% ([Table 4]). The Hosmer-Lemeshow test confirmed a good model fit (p = 0.160). Adjusting for time since surgery did not alter the model's significance (Cox & Snell $R^2$ = 0.239, $\chi^2$ = 15.569; p = 0.001; classification accuracy: 78.9%). Both ST and PMI remained significant predictors, with a good model fit confirmed by the Hosmer-Lemeshow test (p = 0.321). For each one-point increase in ST, the risk of developing HLNP increased by 1.056 times, while each one-point increase in PMI reduced the risk of HLNP by 0.204 times. The post-hoc power analysis indicated power as 0.05 (OR: 1.056, α = 0.05, n = 57, probability of Y = 0.38) for ST. For PMI, the power reached 0.99 (OR: 4.9, calculated as 1/0.204). Similar to the univariate regression results, both factors require careful interpretation due to their modest ORs and wide confidence intervals.

When a PDQ score ≥ 19 was used as the dependent variable, no variables met significance in the initial multivariate logistic regression. However, after adjusting for time since surgery, PMI emerged as a statistically significant predictor (p = 0.048), although the overall model significance was marginal (Cox & Snell $R^2$ = 0.081, $\chi^2$ = 4.784, p = 0.091). The Hosmer-Lemeshow test confirmed a good fit (p = 0.512) and with a classification accuracy of 86%. Post-hoc analysis reported a power of 0.99 with an OR of 8.62 (1/0.116), α = 0.05, sample size = 57, and probability of Y = 0.14.

ROC curve analysis identified optimal cutoff scores for ST and PMI in predicting HLNP ([Fig 3]). The optimal cutoff score for ST was 23.81, with an AUC of 0.722, sensitivity of 0.800, and specificity of 0.591, suggesting that patients with ST scores above this threshold are more likely to develop HLNP. The cutoff for PMI was 9.82, yielding an AUC of 0.698, sensitivity of 0.743, and specificity of 0.636, indicating that patients with PMI values > 9.82 are more likely to remain free of HLNP. When PMI was analyzed using the alternative PDQ score ≥ 19, the cutoff value was 9.755, with an AUC of 0.695 (p = 0.079), sensitivity of 0.776 and specificity of 0.625. All AUC values met acceptable diagnostic thresholds.

### Factors correlated with the PMI and ST

Spearman's rank correlation assessed relationships between the number of resected lymph nodes and physical variables ([Table 5]). In BCS who underwent sentinel lymph node biopsy, an increased number of resected lymph nodes was associated with reduced external rotation and decreased PMI. The type of breast reconstruction and adjuvant therapy also influenced these associations. ST scores were statistically significantly higher in patients who underwent chemotherapy compared to those who did not (median: 23.81 vs. 14.29, p = 0.017). ST demonstrated statistically significantly correlations

**Table 4. Multivariate logistic regression of factors associated with homolateral neuropathic pain.**

| PDQ ≥ 13 | Crude analysis | | Adjusted analysis[a] | |
|---|---|---|---|---|
| Predictors | OR (*p*-value) | 95% CI | OR (*p*-value) | 95% CI |
| ST | 1.055 (0.007)** | 1.015–1.097 | 1.056 (0.007)** | 1.015–1.098 |
| PMI | 0.191 (0.033)* | 0.042–0.877 | 0.204 (0.047)* | 0.043–0.977 |
| PDQ ≥ 19 | Crude analysis | | Adjusted analysis[a] | |
| Predictors | OR (*p*-value) | 95% CI | OR (*p*-value) | 95% CI |
| PMI | No variable selected | | 0.116 (0.048)* | 0.014–0.980 |

[a]Performed by time since surgery to enrollment. PDQ, painDETECT questionnaire score; NP, neuropathic pain; OR, odds ratio; ST, systemic therapy side effect; PMI, pectoralis minor length index.

*and

**indicate *p*-value < 0.05 and 0.01, respectively.

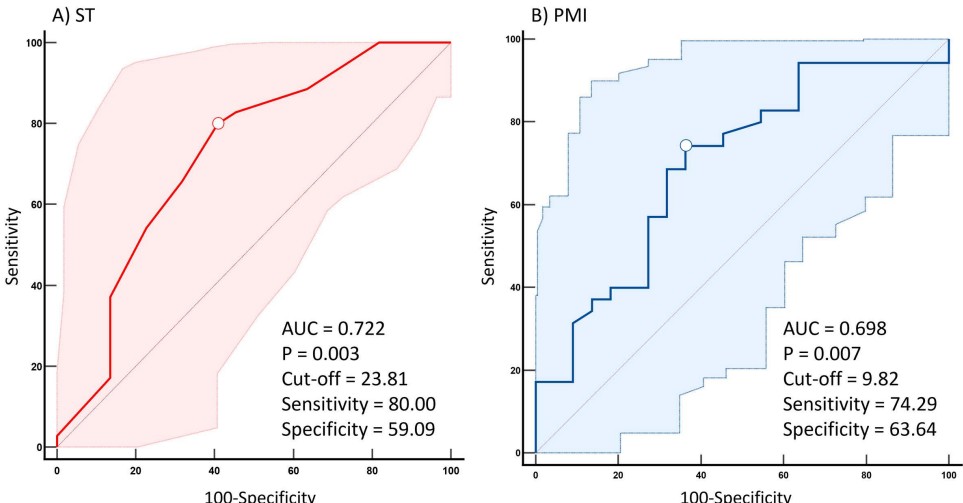

**Fig 3. Cut-off values of systemic therapy side effects and pectoralis minor length index for predicting homolateral neuropathic pain.** (A) The cut-off score for systemic therapy side effects (ST) was 23.81, indicating that patients with an ST score above 23.81 are more likely to develop HLNP (AUC = 0.722). (B) The cut-off value for the pectoralis minor length index (PMI) was 9.82 cm/cm, indicating that patients with a PMI greater than 9.82 are more likely to be free of HLNP. (AUC = 0.698).

**Table 5. Correlation between the number of lymph nodes resected and shoulder mobility at 1-month postoperative.**

| Group | | N | Flexion | Abduction | External rotation | Internal rotation | PMI |
|---|---|---|---|---|---|---|---|
| All | | 57 | 0.005 (0.485) | −0.097 (0.238) | 0.296* (0.013) | −0.080 (0.277) | −0.203 (0.065) |
| Type of reconstruction | ABR | 25 | −0.067 (0.375) | −0.179 (0.196) | 0.259 (0.105) | −0.155 (0.230) | −0.080 (0.352) |
| | IBR | 32 | 0.072 (0.347) | −0.016 (0.465) | 0.299* (0.048) | −0.032 (0.431) | −0.349* (0.025) |
| Type of lymph node dissection | SLNB only | 44 | −0.076 (0.313) | −0.068 (0.332) | 0.330* (0.014) | 0.141 (0.181) | −0.317* (0.018) |
| | ALND only | 5 | −0.564 (0.161) | −0.718 (0.086) | 0.051 (0.467) | −0.718 (0.086) | −0.051 (0.467) |
| | Both | 8 | 0.119 (0.389) | 0.048 (0.455) | 0.190 (0.326) | −0.024 (0.478) | −0.143 (0.368) |
| History of chemotherapy | No | 28 | 0.065 (0.372) | 0.091 (0.323) | 0.462** (0.007) | −0.048 (0.405) | −0.286 (0.070) |
| | Yes | 29 | 0.019 (0.460) | −0.159 (0.205) | 0.054 (0.391) | −0.198 (0.151) | −0.071 (0.358) |
| History of radiation therapy | No | 41 | −0.051 (0.376) | −0.144 (0.184) | 0.351* (0.012) | −0.009 (0.477) | −0.074 (0.322) |
| | Yes | 16 | 0.284 (0.144) | 0.111 (0.341) | 0.334 (0.103) | −0.344 (0.096) | −0.548* (0.014) |
| History of hormone therapy | No | 20 | −0.001 (0.499) | −0.130 (0.293) | 0.712** (< 0.001) | −0.336 (0.074) | 0.043 (0.429) |
| | Yes | 37 | −0.026 (0.439) | −0.071 (0.338) | −0.015 (0.464) | 0.155 (0.180) | −0.345* (0.018) |

Data are shown in Spearman's rho (*p*-value). As the directional hypothesis was established, a one-tailed Spearman's rank test was conducted. PMI, pectoralis minor length index; ABR, abdominally based reconstruction; IBR, implant-based reconstruction; SLNB, sentinel lymph node biopsy; ALND, axillary lymph node dissection.

*and

**indicate *p*-value < 0.05 and <0.01, respectively.

with insomnia (ρ = 0.602, p < 0.001), fatigue (ρ = 0.559, p < 0.001), arm symptoms (ρ = 0.430, p = 0.001), pain (ρ = 0.329, p = 0.012), and body image (ρ = −0.327, p = 0.013). Furthermore, PMI was positively correlated with shoulder abduction (ρ = 0.297, p = 0.013) and negatively correlated with arm symptoms (ρ = −0.353, p = 0.004) and average pain intensity over the preceding month (ρ = −0.279, p = 0.018). PMI was higher in BCS who underwent sentinel lymph node biopsy (N = 52) compared with those who underwent only axillary lymph node dissection (N = 5); however, the difference was not statistically significant (median: 9.98 vs. 9.80, p = 0.337). Correlation coefficients between these variables are provided in S2 Table.

## Discussion

In this study, we investigated the predictive value of physical parameters, specifically the PMI and ST, for developing HLNP during recovery following mastectomy with lymph node resection. Furthermore, we explored the impact of lymph node resection on shoulder mobility. Our findings revealed that early evaluation of PMI and ST can effectively provide valuable predictive insight into HLNP risk, while the extent of lymph node resection may contribute to pectoralis minor muscle shortening, especially in patients undergoing breast reconstruction and adjuvant therapies. PMI can be readily measured using specific instruments or calipers, while ST can be assessed using global questionnaires.

Our findings deviate from prior research that identified acute postoperative pain as a direct predictor of NP [11,15,48]. While correlations were observed between pain intensity, ST, and PMI, our results suggest that musculoskeletal changes and ST indirectly influence HLNP development. These discrepancies from previous studies may stem from methodological differences, such as the exclusion of psychological assessments and our study's younger, surgically distinct cohort [8–10,12–14].

Despite the wide confidence interval of PMI in both HLNP models, we consider PMI a clinically and statistically relevant indicator of HLNP prevalence. The results indicate that higher preceding pain intensity is associated with a lower PMI, while a higher PMI correlates with greater shoulder abduction. Pain intensity was omitted from the final analysis; however, these findings underscore the role of acute pain in PMI changes and HLNP development. Given the weak correlations between PMI and current pain intensity, we propose that BCS adopt protective slouched postures due to acute postoperative pain [21]. Considering the wide confidence interval of PMI, a lower PMI, as a reaction to pain, may serve as a preceding factor for HLNP presence. However, further validation with larger sample sizes is necessary to confirm these associations.

Biomechanically, a reduced PMI indicates scapular depression and protraction, restricting pectoralis major muscle mobility during arm elevation [17]. These biomechanical alterations may induce abnormal movement sensations and soft tissue injury around the shoulder [21]. Chronic pain development in this context likely involves central sensitization, while scapular malalignment exacerbates peripheral sensitization through pectoral nerve compression [23]. Furthermore, scapular malalignment has been linked to increased local pain pressure sensitivity [22], a hallmark of peripheral sensitization. These findings corroborate prior studies linking chest tightness and upper limb discomfort [20], supporting scapular dysfunction as a risk factor for HLNP. Since we did not measure sensitization properties, this hypothetical interpretation calls for further studies that use specific measurements such as quantitative sensory testing.

In addition to analgesic prescription, early initiation of ROM exercises and muscle stretching may mitigate HLNP development [49]. Despite the statistically insignificant differences, prevalence of HLNP decreased by 14%. We believe that our rehabilitation partially contributed to the reduction of HLNP prevalence during follow-up. A prior study initiated free ROM exercises 15 days postoperatively [50], whereas participants in this study began stretching around 30 days postoperatively. Pain intensity and Quick–DASH scores at baseline in our cohort were comparable to participants in the control group of a similar study initiating exercises at 30 days. As moderate-to-severe acute pain following surgery has been related with a three-fold increased risk of NP development [11], earlier intervention may alleviate pain intensity and improve ROM. In addition, scapular alignment exercises utilizing biofeedback modalities can enhance shoulder function

by addressing mechanical sensitivity [51] and pain [52]. Manual therapy has demonstrated efficacy in reducing pain post-breast cancer therapy [53] through its physiological and biomechanical effects on musculature and the peripheral nervous system. Based on our results and previous literature, early multifaceted physical therapy may help prevent HLNP.

AUC analysis identified a cut-off value of 9.82 for predicting HLNP compared with the PMI value of 10 observed in healthy participants [33], indicative of pectoralis minor muscle shortness. Furthermore, the cutoff value decreased to 9.755 when HLNP was defined using a PDQ score of ≥19, reinforcing its clinical relevance. Thus, we consider the cutoff value to be clinically significant.

ST was statistically significant; however, its modest odds ratio (1.056) and low statistical power suggest that its predictive strength is limited. Therefore, its association with insomnia, fatigue, and arm symptoms indicates that it may play an indirect role in HLNP development. While no direct correlation was found between ST score and pain severity, ST was statistically significantly associated with symptoms such as insomnia, fatigue, arm symptoms, pain, and body image subscale scores from the questionnaires. Given the significant interconnection between insomnia, fatigue, and pain [54], and the negative impact of pain—including arm symptoms—on body image [55], a higher ST score may precede pain chronification. Since chemotherapy is the primary factor influencing ST scores, particular attention should be given to the side effects experienced by BCS who have undergone chemotherapy.

Another novel observation was the interaction between the amount of resected lymph nodes and physical variables influenced by reconstruction type and adjuvant therapy. Our results partially support the hypothesis of a significant association between the number of lymph node resections and PMI. Although the difference was not statistically significant due to the imbalance in sample size (52 vs. 5), PMI was lower in BCS who underwent only axillary lymph node dissection compared with those who had a sentinel lymph node biopsy. Furthermore, the correlation test for the sentinel lymph node biopsy group supports this hypothesis. Given that the axillary lymph nodes are anatomically located near the pectoralis minor muscle [26], the extent of axillary surgery may contribute to pectoral minor muscle length changes, potentially affecting shoulder mobility and upper limb function. These findings highlight the clinical importance of postoperative musculoskeletal assessment in BCS undergoing axillary surgery.

Surgical differences appear to significantly influence this relationship. Implant-based reconstruction, requiring significant soft tissue excision, muscle detachment, and material anchoring, could intensify postoperative pain and prolong pectoral minor muscle recovery [56]. Conversely, transverse rectus abdominis flap reconstruction prevents abnormal muscle activation patterns as this approach does not require significant soft tissue excision and preserves the pectoral major muscle. Instead, it fills the empty spaces with donor-site adipose tissue, potentially reducing muscular tightness and postoperative discomfort [24].

However, correlation coefficients for external rotation contradicted previous findings [25], suggesting that reconstruction type and history of adjuvant therapy modulate these relationships differently. Lymph node resection may compromise anterior shoulder integrity due to axillary soft tissue excision, creating empty spaces that reduce muscular force in shoulder internal rotators [57]. Radiotherapy significantly contributes to pectoral tightness [21] and recovery limitations in shoulder abduction [19], with its effects on soft tissue fibrosis influenced by the extent of axillary surgery [58]. Hormone therapy-induced joint pain may exacerbate PMI shortening. These findings emphasize the need for preoperative multidisciplinary collaboration to align reconstruction type, pharmacological therapy, timing and components of assessments, and physical therapy goals, thereby reducing HLNP risk.

The strength of this study lies in its practical applicability, particularly for identifying NP during the acute postoperative phase. However, it has some limitations that may affect the broader applicability of our findings. First, the relatively small sample size and imbalance in surgical data may limit generalizability. The small sample size contributed to the wide confidence interval for the OR of PMI and the significant but very low OR of ST, both of which reduced the statistical power of our analyses. Owing to the small sample size, the number of variables included in the model was also limited. To address this, we implemented a two-step logistic regression to identify effective predictors and covariates, during which ordinal

variables, such as patient age, were excluded as part of the model selection strategy. Second, sole reliance on questionnaires for HLNP classification may undermine diagnostic robustness; incorporating quantitative sensory testing could improve diagnostic precision. Third, while PMI measurement demonstrated excellent inter-tester reliability (ICC = 0.945) [31], its convenience may compromise accuracy compared to radiographic methods. Due to the lack of existing literature, a normative PMI cutoff value of 10 in young, asymptomatic individuals was referenced [33]. However, this comparison should be re-evaluated once a standardized PMI cutoff for the operated side in BCS is established. Fourth, the study did not include data on the contralateral side, preoperative physical assessments, and psychological evaluations of anxiety and depression, which are critical for comprehensive HLNP analysis. Finally, there was a limitation regarding measurements. We did not assess sensitization, which reduces the strength of our interpretation. Consequently, we could not fully evaluate the underlying rationale for how pectoralis minor muscle tightness may contribute to HLNP development. Furthermore, some questions relied on the patient's memory, potentially introducing recall bias.

## Conclusion

This study identified the PMI and ST at 1 month postoperatively as significant predictors of HLNP at 4 months following mastectomy with lymph node resection. PMI < 9.82 and ST > 23.81 were associated with an elevated risk of HLNP, a condition significantly correlated with upper limb functional impairment. Both predictors were significantly correlated with acute pain and its chronification. These findings underscore the critical need for early postoperative evaluations to identify patients at heightened risk for NP and associated functional limitations. Multidisciplinary postoperative care plans incorporating comprehensive symptom assessments targeted physical interventions, and close monitoring within the first-month post-surgery are strongly recommended to mitigate HLNP risk. Future research should focus on larger cohorts and randomized controlled trials to validate these findings by comparing the effects of early initiation of physical therapy to increase PMI versus usual care.

## Supporting information

**S1 Appendix. Exercise description.**
(PDF)

**S2 Table. Correlation coefficients between variables.**
(XLSX)

**S3 Checklist. STROBE checklist.**
(DOCX)

**S4 Checklist. PLOSOne human participants research checklist.**
(DOCX)

## Acknowledgments

We wish to thank all the patients, caregivers, and staff from all units that participated in the study.

## Author contributions

**Conceptualization:** Asall Kim, Chunghwi Yi, Jaewon Beom.

**Data curation:** Asall Kim, Myungki Ji, Jae-Young Lim, Yujin Myung, Eun Joo Choi, Hee-Chul Shin, Jaewon Beom.

**Formal analysis:** Asall Kim, Chunghwi Yi, Ui-Jae Hwang, Jaewon Beom.

**Investigation:** Asall Kim, Chunghwi Yi, Jaewon Beom.

**Methodology:** Asall Kim, Chunghwi Yi, Ui-Jae Hwang, Jaewon Beom.

**Project administration:** Jaewon Beom.

**Resources:** Myungki Ji, Jae-Young Lim, Yujin Myung, Eun Joo Choi, Hee-Chul Shin, Jaewon Beom.

**Supervision:** Jaewon Beom.

**Validation:** Jaewon Beom.

**Visualization:** Asall Kim, Eun Joo Choi, Jaewon Beom.

**Writing – original draft:** Asall Kim.

**Writing – review & editing:** Chunghwi Yi, Myungki Ji, Ui-Jae Hwang, Jae-Young Lim, Yujin Myung, Eun Joo Choi, Hee-Chul Shin, Jaewon Beom.

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
