## [Decision Letter · Decision Letter 0]

Dear Dr. Beom,

We look forward to receiving your revised manuscript.

Kind regards,

Muhammad Shawqi, MD, MSc

Guest Editor

PLOS ONE

Journal Requirements:

2. We note that there is identifying data in the Supporting Information file <study data>. Due to the inclusion of these potentially identifying data, we have removed this file from your file inventory. Prior to sharing human research participant data, authors should consult with an ethics committee to ensure data are shared in accordance with participant consent and all applicable local laws.

-Location data

Additional guidance on preparing raw data for publication can be found in our Data Policy (https://journals.plos.org/plosone/s/data-availability#loc-human-research-participant-data-and-other-sensitive-data ) and in the following article: http://www.bmj.com/content/340/bmj.c181.long .

Please remove or anonymize all personal information (ID), ensure that the data shared are in accordance with participant consent, and re-upload a fully anonymized data set. Please note that spreadsheet columns with personal information must be removed and not hidden as all hidden columns will appear in the published file.

4. Please include captions for your Supporting Information files at the end of your manuscript, and update any in-text citations to match accordingly. Please see our Supporting Information guidelines for more information: http://journals.plos.org/plosone/s/supporting-information .

Additional Editor Comments:

- Please ensure that the English version of the IRB is signed by the chairperson and clearly stating the start and end dates of the approval effective period in a similar way to the Korean version of the IRB (from 2021-07-29 to 2022-07-28). In addition, the date next to the missing signature should be the same or after the approval date (2021-09-07).

- Please ensure that the study registration status (+/- the registration platform and the registration number) is clearly stated in the methodology section.

- Please ensure that a well-explained participant flow-chart is included as a diagram or figure in the methodology section.

- Please ensure that all references in the list of references follow the Vancouver styling and are properly formatted.

Reviewers' comments:

Reviewer's Responses to Questions

**Comments to the Author**

1. Is the manuscript technically sound, and do the data support the conclusions?

Reviewer #1: Yes

Reviewer #2: Yes

Reviewer #3: Partly

2. Has the statistical analysis been performed appropriately and rigorously?

Reviewer #1: Yes

Reviewer #2: Yes

Reviewer #3: Yes

3. Have the authors made all data underlying the findings in their manuscript fully available?

Reviewer #1: Yes

Reviewer #2: Yes

Reviewer #3: Yes

4. Is the manuscript presented in an intelligible fashion and written in standard English?

Reviewer #1: Yes

Reviewer #2: Yes

Reviewer #3: Yes

Reviewer #1: This prospective observational study aimed to determine whether changes in the pectoral muscle after breast cancer surgery significantly predict the development of neuropathic pain (NP). Among the 57 patients who underwent breast cancer surgery, reconstruction, and lymph node dissection, 22 patients developed NP four months later (as defined by a PainDetect score exceeding 13). Multivariate logistic regression analysis revealed that systemic therapy side effects (ST) and the pectoralis minor length index (PMI) at one month after surgery were significant predictors of NP at four months. Additionally, an increased number of resected lymph nodes was associated with decreased PMI. ST scores were statistically significantly higher in patients who underwent chemotherapy compared to those who did not.

The reviewer personally believes that the factors are likely complex, but this study provides very interesting results that suggest necessary interventions after surgery. There are concerns outlined below.

Major Concerns:

1. Although 135 patients were screened, more than half were excluded. The reasons for this should be stated.

2. It appears that rehabilitation started one month after surgery, but what kind of pain relief medication was prescribed at the one-month mark?

3. This study investigates the prediction of NP four months after surgery based on data from one month post-surgery. What kind of rehabilitation program was implemented for the subjects during those three months? Also, was any pain relief treatment administered during those three months? These are important pieces of information in this study focused on NP.

Minor Concerns: On page 23, it states, "Given the role of acute pain in NP development [16], the observed inconsistencies between our univariate and multivariate models are plausible." The reviewer does not understand this meaning. Could the differences between univariate and multivariate analysis results simply be due to sample size issues?

Reviewer #2: The manuscript presents a novel and clinically relevant study with a strong methodological foundation. However, minor revisions are needed to improve readability, structure, and clarity of statistical considerations. Addressing these aspects will enhance the overall impact and acceptance potential of the study.

• Strengths:

o The abstract is well-structured and clearly outlines objectives, methods, key results, and clinical implications.

o The study uses appropriate statistical tests, including logistic regression and ROC curve analysis. The presentation of odds ratios (OR), confidence intervals (CI), and p-values adds credibility.

o Generally clear and well-written with minor grammatical inconsistencies.

o Technical terms are used correctly with appropriate referencing.

• Weaknesses:

o A flowchart should be added to the manuscript to enhance clarity and provide a visual summary of the study methodology

o The sample size (N=57) was relatively small, which may limit generalizability and statistical power.

o Multivariate analysis only considers variables with p < 0.10 in univariate analysis, which might overlook potential confounding factors.

o No power analysis was reported to determine if the sample size is sufficient for detecting meaningful differences.

Reviewer #3: Thank you for the opportunity to review this interesting paper. However, there are several concerns that should be addressed. I hope the following comments will help enhance the study's impact and improve its readability.

INTRODUCTION:

The introduction predominantly relies on speculative links between biomechanics, psychological factors, and central sensitization. However, the study does not assess psychological or sensitization factors. I recommend a major revision of this section to focus more closely on the study's specific outcomes.

Abbreviations: Please reduce the number of abbreviations, as the excessive use hinders readability.

Line 82: Specify whether you are referring to the pectoralis minor or major muscle.

Line 85: The hypothesis regarding lymph nodes is not supported by the introduction.

Lines 117–118: Provide more details about the primary outcomes, including their psychometric properties between days.

Lines 119–120: In fact, the measurement used for PMI targets to estimate the distance between PM muscle origin and insertion and not scapular position. Please clarify.

Line 122: Why was this measured in the supine position, while the study mentions it was done standing?

Line 123: Update reference 34 with the appropriate citation: Borstad JD, Ludewig PM. The effect of long versus short pectoralis minor resting length on scapular kinematics in healthy individuals. J Orthop Sports Phys Ther. 2005;35(4):227-38.

-METHODS

- Line 95: Was there no age limit for participants? If so, why?

Line 96: Clarify the recruitment setting.

Lines 114–115: Add more details about the intervention and ensure that the discussion section uses the treatment delivered in this study rather than referencing previous studies with different techniques (see lines 330–333).

Lines 129–130: As primary outcomes, enough details should be provided in the manuscript itself rather than directing readers to previous studies.

Line 142: Relying on patient memory to rate the strongest pain over a four-week interval introduces bias, as patient recall is not a reliable tool, especially over time. Please address this in the limitations section. Table 2 also suggests potential bias in the HLNP group (score = 5.5, only for this measurement).

Line 153: The abbreviation "Quick-DASH" is more commonly used than "Q-DASH."

Line 154: The responsiveness demonstrated in study 44 has not been validated in a cancer population; please address this.

Statistical analysis: The manuscript lacks details regarding sample size estimation.

Line 161: Clarify which groups are being referenced. Based on the results, I assume the groups were defined by neuropathic pain, but this is unclear in the methods section.

RESULTS

Line 185: Why is this data important for the study population? Would patients with diabetes have been excluded?

Table 2: Clarify why the Non-HLNP group presented more patients with pectoral tightness but less pectoralis minor shortening, as well as why this group demonstrated 16 degrees more abduction ROM compared to the HLNP group. This requires further explanation in the discussion.

Lines 212–219: The odds ratios (ORs) reported are close to 1, with confidence intervals that cross 1 or are very wide (e.g., CI = 0.046–0.74). While these results are statistically significant, are they clinically meaningful? Please integrate this discussion into the main findings and conclusions.

Lines 118–119: Why did you decide to present this result? The study's primary focus was on a PDQ score ≥ 13. Please clarify.

Table 3: Why present both cutoff values? The primary focus was on a PDQ score ≥ 13.

Table 3: Highlight the wide confidence intervals for PMI in the discussion.

Table 3: The systemic therapy side effect results were modest. Are these findings clinically important?

Lines 235–236: Again, discuss the discrete results and large confidence intervals (Table 4) in the discussion.

Line 237: Add this analysis to the statistical analysis section.

Lines 240–241: The small sample size is a study limitation. The reader needs to have clarity that the results were not large and that the sample was small for the number of variables included in the regression analysis.

Lines 250–257: Congratulations on adding this analysis—it is practical for clinicians and future research. Well done!

DISCUSSION

The discussion focuses heavily on sensitivity and sensitization. Why were these not assessed in the study? This should be acknowledged as a limitation.

Lines 287–289: Is ST considered a physical parameter? Based on Table 2, side effects seem to have influenced HLNP. How could this be evaluated earlier?

Lines 310–312: This sentence does not align with the paragraph's context and requires further development.

Lines 313–314: Be cautious with your claims; the study did not assess the pectoral fascia.

Lines 342–344: The transitions are abrupt, and this sentence is underdeveloped. Expand on this idea and connect it to your findings.

Lines 350–355: Further discussion is needed. Instead of simply comparing your results with previous studies, emphasize the clinical impact and relevance of your findings.

CONCLUSION

Lines 384–386: The results do not substantiate this speculation.

Line 389: Clarify how RCTs could validate these results.

Line 390: The type of study conducted does not support this claim.

**Do you want your identity to be public for this peer review?** For information about this choice, including consent withdrawal, please see our Privacy Policy

Reviewer #1: No

Reviewer #2: No

Reviewer #3: No

---

## [Author Response · Author response to Decision Letter 1]

10 Apr 2025

The response to reviewers is provided in a separate file labeled "Response to Reviewers.docx."

---

## [Decision Letter · Decision Letter 1]

Dear Dr. Beom,

In the meantime, I may send you one further peer review toensure that overstated points in the paragraph 3 will be rewritten in a more skeptical tone, mainly regarding:

1- how PM changes post-breast cancer surgery might influence neuropathic pain development because (reference 22) does not mention that scapular depression is not a proven consequence of PM tightness.

2- There is no established cutoff value for PM tightness in clinical populations. Therefore, this a must written limitation that the authors must state to justify the use of values from young, asymptomatic individuals (reference 18). Also, this point needs to be stated again in the limitations section in details

We look forward to receiving your revised manuscript.

Kind regards,

Muhammad Shawqi, MD, MSc

Guest Editor

PLOS ONE

Journal Requirements:

Please review your reference list to ensure that it is complete and correct. Any changes to the reference list should be mentioned in the rebuttal letter that accompanies your revised manuscript. If you need to cite a retracted article, indicate the article’s retracted status in the References list and also include a citation and full reference for the retraction notice.

Additional Editor Comments (if provided):

Thank you for submitting this revised manuscript. To sum up, more coherence is needed in the paragraph 3 to abolish any overstating arguments not supported by a valid reference. It would be more preferable that the tone of authors should be clear, accurate and limitation-oriented rather than overstating.

Reviewers' comments:

Reviewer's Responses to Questions

**Comments to the Author**

Reviewer #1: All comments have been addressed

Reviewer #2: All comments have been addressed

Reviewer #3: (No Response)

2. Is the manuscript technically sound, and do the data support the conclusions?

Reviewer #1: Yes

Reviewer #2: Yes

Reviewer #3: (No Response)

3. Has the statistical analysis been performed appropriately and rigorously?

Reviewer #1: Yes

Reviewer #2: Yes

Reviewer #3: (No Response)

4. Have the authors made all data underlying the findings in their manuscript fully available?

Reviewer #1: Yes

Reviewer #2: Yes

Reviewer #3: (No Response)

5. Is the manuscript presented in an intelligible fashion and written in standard English?

Reviewer #1: Yes

Reviewer #2: Yes

Reviewer #3: (No Response)

Reviewer #1: Thank you for addressing the reviewer's comments.

This manuscript has become clearer and I believe it has improved. I have no further additional comments.

Reviewer #2: All reviewer comments have been thoroughly addressed, and the authors have carefully implemented the necessary revisions throughout the manuscript. These changes have significantly improved the clarity, coherence, and overall scientific quality of the paper. I believe that the revised version is now suitable for publication and meets the journal’s standards.

Reviewer #3: (No Response)

**Do you want your identity to be public for this peer review?** For information about this choice, including consent withdrawal, please see our Privacy Policy

Reviewer #1: No

Reviewer #2: No

Reviewer #3: No

---

## [Author Response · Author response to Decision Letter 2]

22 May 2025

Please review the response to the reviewer file.

---

## [Editor Report · Decision Letter 2]

Pectoralis minor length index at 1 month postoperative can predict homolateral neuropathic pain 4 months after mastectomy with lymph node resection

PONE-D-24-55787R2

Dear Dr. Beom,

We’re pleased to inform you that your manuscript has been judged scientifically suitable for publication and will be formally accepted for publication once it meets all outstanding technical requirements.

Kind regards,

Muhammad Shawqi, MD, MSc

Guest Editor

PLOS ONE

Additional Editor Comments (optional):

Thank you to all the study team for their innovative work. All raised concerns have now been adequately addressed.

---

## [Editor Report · Acceptance letter]

PONE-D-24-55787R2

PLOS ONE

Dear Dr. Beom,

I'm pleased to inform you that your manuscript has been deemed suitable for publication in PLOS ONE. Congratulations! Your manuscript is now being handed over to our production team.

Kind regards,

on behalf of

Dr. Muhammad Shawqi

Guest Editor

PLOS ONE